# Numerical Analysis of Blast Effects and Mitigation in the Far-Field from Small Explosions

Adam G. Taylor

Lawrence Livermore National Laboratory, Computational Engineering Division, Livermore, CA 94550, USA; taylor265@llnl.gov

**Abstract:** Requirements for explosive safety are often given in terms of a "*K*-Factor", correlating incident blast effects with the distance and TNT equivalent weight of an explosion. Traditionally, this is conducted using empirical correlations to experimental measurements (e.g., the Kingery–Bulmash equations). In the far-field, empirical verification of incident overpressure and impulse magnitudes can be difficult; extrapolations from data give expected values at reasonable standoff distances that sometimes are too small to measure on available equipment but are larger than some regulations require. The present paper describes the results of numerical hydrocode analysis to verify the expected incident overpressure and impulse from small hemispherical ground charges of TNT at these relatively large distances. Furthermore, the dynamic effect of incident blast waves on lightweight, modular mitigation barriers is studied to gauge their effectiveness at providing safety standard compliance.

**Keywords:** blast effects; mitigation; hydrocode analysis

## 1. Introduction

Understanding the structure of blast waves and the dynamics of their interactions with structures is key for mitigation and safety. The formation and propagation of these shock waves is a highly nonlinear dynamic process; thus, prediction of the incident waveforms and their corresponding blast overpressure and impulsive loads for a given scenario can be difficult. It is common for various government, military, and scientific institutions to prescribe criterion for safety from blast effects in terms of "*K* factors":

$$K = R/W^{1/3} \tag{1}$$

Here, $R$ is the distance from the explosive source and $W$ is the net explosive (TNT equivalent) weight. Allowable exposure for personnel, nearby structures, and withdrawal distances can be given in terms of these $K$ factors, which have been empirically correlated to values of incident overpressure and impulse. A figure regularly encountered in explosive safety documentation is the K328 criterion, often referred to as the "Public Withdrawal Distance"; calculated using units of ft/lb$^{1/3}$ this corresponds to a peak incident overpressure of 0.0655 psi, (0.4516 kPa) and is said to be a condition under which there is no probability of harm. Different safety guidelines have different requirements for personnel, but they are very commonly given in terms of these $K$ factors.

The empirical nexus of $K$ factor correlation appears to be the work of Kingery [1,2]. The original data came from quite large (5, 20, 100, and 500 ton) hemispherical TNT events. Instrumentation at various distances measured arrival time, peak overpressure, and the duration of the positive pressure phase and the positive impulse. This same data set was later reinterpreted and extrapolated to include reflected pressures/impulse and shock velocities by Kingery and Bulmash [3]. It is these fits which became the basis for more widespread application, and thus the empirical equations are often referred to as Kingerly-Bulmash (KB) curves. Swisdak [4] provides a good overview of this history, along with

improved equations fitting the same data. More recent fits by Jeon et al. [5] claim to further simplify the curves with the same accuracy.

There is definite uncertainly in the accuracy of the KB curves and other analytic and empirical tools for predicting blast overpressure in a given case. Karlos et al. [6–8] have investigated the structure of blast waves and their parameters for scaling and decay, including variations in explosive type, weight, and configuration on the resulting incident pulses. A recent review of analytical and empirical prediction methods by Ullah et al. [9] shows a very large spread in the predicted blast overpressures and wave structures from various accepted sources. Recent repeated blast measurements from Stewart et al. [10] show large variability in the measured results from what are ostensibly the same experiments. The recent experiment of Filice et al. [11] provides more data and KB comparisons and variances for relatively nearby (2–5 m) and relatively small (100–400 g) explosives. In a review of the experimental literature vs. KB predictions, Rigby et al. [12] state that the variation in experimental predictions is so large in nominally similar experiments that there is a valid question as to whether blast phenomena are inherently deterministic, or whether they should be viewed as fundamentally stochastic processes. Under this lens, KB and others may be viewed as useful only at predicting the order of magnitude of blast effects.

The question arises: can direct physics-based calculation of blast wave parameters provide more detailed and accurate predictions for a given case of interest?

The classical analytical result for the prediction of the evolution of a very strong explosion is the so-called Taylor–von Neumann–Sedov solution [13–16]. This applies only to spherical (1D) blasts and is derived under assumptions (point source, zero ambient pressure) that leave it applicable only for intermediate distances. Some of the earliest published attempts to simulate explosions under real conditions (i.e., into non-zero ambient pressure conditions) were performed by Brode [17,18] and Goldstine and von Neumann [19].

More recently there have been various simulations performed in modern software packages aimed at the prediction of the evolution of blast waves. Ding et al. [20,21] recently presented the results of numerical simulations of very large TNT equivalent blasts and their resulting effects on near and far-field structures. Xue et al. [22] modeled the whole process of explosive shockwave formation and propagation from relatively smaller blasts over larger distances. Sung and Chong have produced a fast-running semi-empirical method for the prediction of blast effects behind shielding barriers; this work includes uncertainty estimations when using KB-type charts [23]. Giodo et al. compared empirical and numerical approaches to investigating the effects of free far-field blasts on masonry wall [24]. Vannucci et al. [25] provide analysis of a blast and shock propagation inside a monumental structure. Draganic and Varevac [26] have provided a useful parametric study on the effects of numerical mesh size on the blast wave parameters.

It is easy to imagine situations (involving explosive training, demolitions, etc.) where relatively small explosions (comparable to 1 kg TNT) send overpressure waves towards personnel relatively far away (30–40 m). These blasts are very small compared to the conditions studied in the published literature or in the data informing KB-type predictions, but nevertheless may induce pressures and impulses in excess of safety guidelines (e.g., the Public Withdrawal Distance). Furthermore, the incident overpressure will be far below the ambient atmospheric pressures and will be difficult or impossible to accurately measure using easily available pressure gauges. Given that there are reasons to question the accuracy of KB-type predictions under these circumstances, research is needed to clarify the situation.

The purpose of this paper is to use numerical tools to investigate cases where very small charges produce relatively small incident overpressure at large distances which still exceed the safety guidelines of public withdrawal distance. The goals here are two-fold:

1.  to predict the structure and magnitude of the incident pressure waves in these cases and to compare to the available empirical blast curves;
2.  to investigate the efficacy of lightweight, modular barriers at mitigating incident overpressure waves to the desired levels.

Towards the first goal, free-field explosions of small hemispherical ground TNT charges into air are simulated out to a range of 40 m. Wave profiles obtained from the free-field simulations are subsequently employed as boundary conditions for dynamic wave-structure interaction models which investigate the second goal.

It is noted that a few different sets of units were used in the preparation of this work. Much of the original work conducted in blast load estimation was conducted in English units (ft/lb/ms/psi) (see for example the original Kingery report [1]). For that reason, explosive range operators and field experts tend to think in terms of these units, and regulations often give quantities such as *K* factors in these units. On the other hand, ALE3D hydrocode analyses are traditionally conducted in a special set of units (cm/g/μs/Mbar). The simulations described herein follow in this tradition. For the sake of consistency, all units in this paper will be given in terms of Si units (m/kg/ms/kPa). In some cases, English units will be listed concurrently.

## 2. Materials and Methods

Simulations presented in this work were performed in ALE3D, a multi-physics software package which utilizes an Arbitrary Lagrangian/Eulerian (ALE) numerical scheme [27]. The numerical simulations performed are of two types: (i) free-field explosions of various weights of TNT in air at atmospheric pressure, and (ii) the dynamic interaction of incident blast waves with simple mitigation barriers. The remainder of this section will describe the material models implemented, and provide further details into the setup of each type of simulation.

### 2.1. Material Models

Three material models were employed for the three separate material components simulated in this work, namely the TNT explosive, the surrounding air, and the Lexan structural barrier. Only the TNT and air appear in the free-field simulations, and only the air and Lexan appear in the blast mitigation simulations. For the explosive TNT, a simple Jones–Wilkens–Lee (JWL) equation of state [28] is used:

$$P(v,e) = A\left(1 - \frac{\omega}{R_1 v}\right)\exp(-R_1 v) + B\left(1 - \frac{\omega}{R_2 v}\right)\exp(-R_2 v) + \frac{\omega}{v}e \qquad (2)$$

Here, $P$ is the pressure, $v = V/V_0 = \rho_0/\rho$ the relative volume, and $e$ is the material energy per reference volume. $V, \rho$ are the volume and density, respectively, while $V_0, \rho_0$ are the initial (reference) values of these properties. The parameter $\omega$ is the Grüneisen coefficient; $A, B, R_1$, and $R_2$ are free parameters. $\omega, R_1$ and $R_2$ are dimensionless, while $A$ and $B$ have units of pressure. The parameter values used in simulations for Equation (2) are given in Table 1.

**Table 1.** JWL parameters for TNT.

| $A$ (kPa) | $B$ (kPa) | $R_1$ | $R_2$ | $\rho_0$ (g/cm$^3$) | $\omega$ |
|-----------|-----------|-------|-------|---------------------|----------|
| $3.712 \times 10^8$ | $3.231 \times 10^6$ | 4.150 | 0.950 | 1.630 | 0.30 |

The equation of state of air is given by a simple Gamma-law:

$$P(\rho, e) = (\gamma - 1)\frac{\rho}{\rho_0}e \qquad (3)$$

The only free parameter $\gamma$ is dimensionless and typically has a value of 1.4 for air. The initial (atmospheric) pressure $P_0$ is obtained through Equation (3) by prescribing and initial energy per unit volume:

$$e_0 = \frac{P_0}{\gamma - 1}$$

The parameter values used in simulations for Equation (3) are given in Table 2.

**Table 2.** Gamma law parameters for air.

| $\gamma$ | $\rho_0$ (g/cm$^3$) | $P_0$ (kPa) |
|---|---|---|
| 1.40 | $1.225 \times 10^{-3}$ | $1.0135 \times 10^2$ |

The Lexan mitigation barrier is modeled using a power law constitutive model:

$$\sigma = k(\epsilon_0 + \epsilon)^{y_c}, \tag{4}$$

$$\epsilon_0 = \left(\frac{E}{k}\right)^{\frac{1}{y_c-1}} \tag{5}$$

Here, $\sigma$ is the current yield stress and $\epsilon$ an equivalent plastic strain. $\epsilon_0$ is an initial yield strain determined by parameters $k, E$ and $y_c$. $E$ is a standard Young's modulus with dimensions of pressure, $k$ the yield stress coefficient with dimensions of pressure, and $y_c$ is a dimensionless strain-hardening coefficient. An additional equation of state relates pressure $P$ to the bulk modulus $K$ and the relative volume $v$:

$$P = K\mu \tag{6}$$

Here, $\mu = (1/v) - 1$, and the bulk modulus is derived from the Young's modulus and Poisson ratio $v$:

$$K = \frac{E}{3(1 - 2v)}$$

The values used for Lexan in the present work are given in Table 3. Given the nature of the low pressure incident waves studied in this paper, only small (elastic) deformations of the barrier are expected. Therefore the values used for the barrier material are not expected to have significant effect on the analysis results.

**Table 3.** Power law parameters for Lexan.

| $E$ (kPa) | $k$ (kPa) | $v$ | $\rho_0$ (g/cm$^3$) | $y_c$ |
|---|---|---|---|---|
| $2.344 \times 10^6$ | $1.119 \times 10^5$ | 0.4 | 1.218 | $2.086 \times 10^{-1}$ |

### 2.2. Free-Field Detonation of TNT

The free-field detonation of hemispherical TNT was simulated under 2D axisymmetric conditions. Figure 1 depicts a cartoon of the setup. The $x = 0$ axis is the axis of rotational symmetry, while the $y = 0$ has symmetry boundary conditions which are used to crudely approximate the ground; however, this approximation causes the simulation to be equivalent to a spherical charge of the same radius exploding in air. The air domain extends from the origin to 40 m in the $x$ and $y$ directions. The outer boundaries have three different boundary conditions applied; "pressure continuous" provides ghost nodes external to the boundary which keeps the pressure constant on the other side, which keeps the initially pressurized gas from expanding and depressurizing as soon as the simulation starts. "Non-reflecting" boundary conditions dampen out any reflected incident waves to minimize boundary effects. The "outflow" condition allows material given outbound velocity to leave the domain.

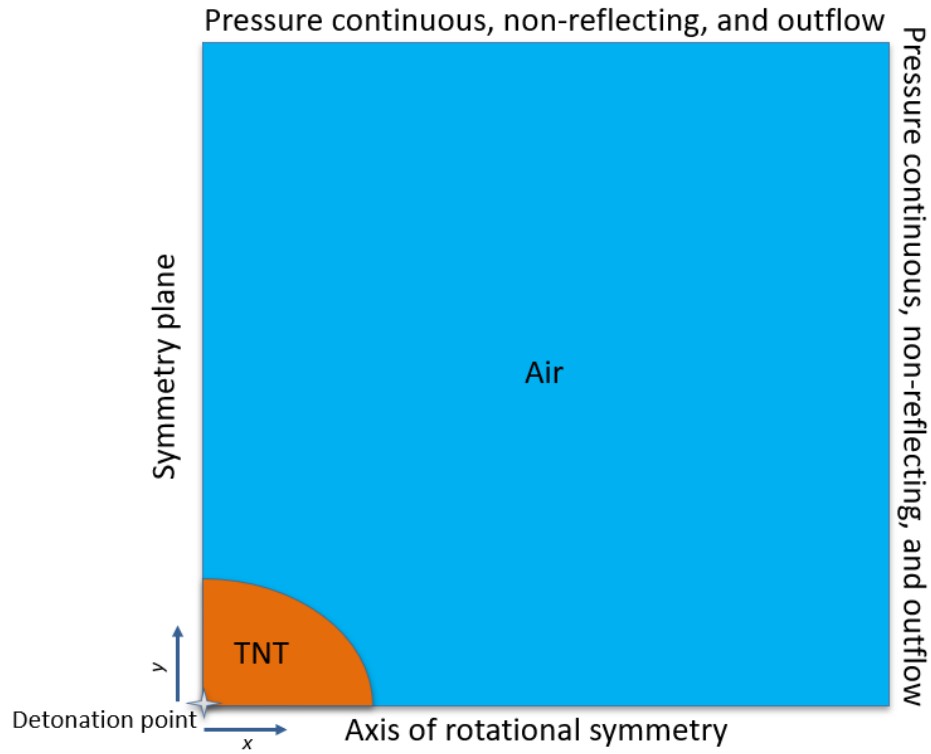

**Figure 1.** A "cartoon" depiction of the setup of the free-field TNT detonation simulations with materials and boundary conditions labeled (Not to scale).

Four simulations in total were performed with of charges with radius 0.0261 m, 0.051 m, 0.0643 m, and 0.081 m, yielding hemispherical charge weights of approximately 0.06123 kg (0.135 lb), 0.45359 kg (1 lb), 0.90718 kg (2 lb), and 1.81436 kg (4 lb), respectively. Note that the charge radius is around 3 orders of magnitude smaller than the domain length; even in 2D, a uniform Cartesian mesh small enough to adequately resolve the TNT would lead to an intractably-large numbers of zones. Instead, a graded mesh approach was use, coarsening with distance from the origin. Initial zone sizes range from approximately $4.5 \times 10^{-3}$ m at the center of the charge out to $5.4 \times 10^{-2}$ m at the outer edge of the domain. The simulations ultimately contained around 7.1 million zones.

Figure 2 shows representative temporal snapshots of pressure in the system as the explosive wave propagates in air. The peak overpressure occurs near the wavefront but rapidly decreases to the ambient pressure and then dips below it for some time before returning. The magnitude of this peak pressure decreases as the wave propagates further from the source. Fixed (Eulerian) pressure tracers were placed every 2 m in the domain just off the $y$-axis in order to study the structure and evolution of the blast wave. Figure 3 shows the results of these pressure tracer time histories. Each tracer shows a pronounced positive overpressure phase followed by a negative phase where pressure dips below ambient. The effects of these negative pressure phases have been studied and are in general not negligible [29,30].

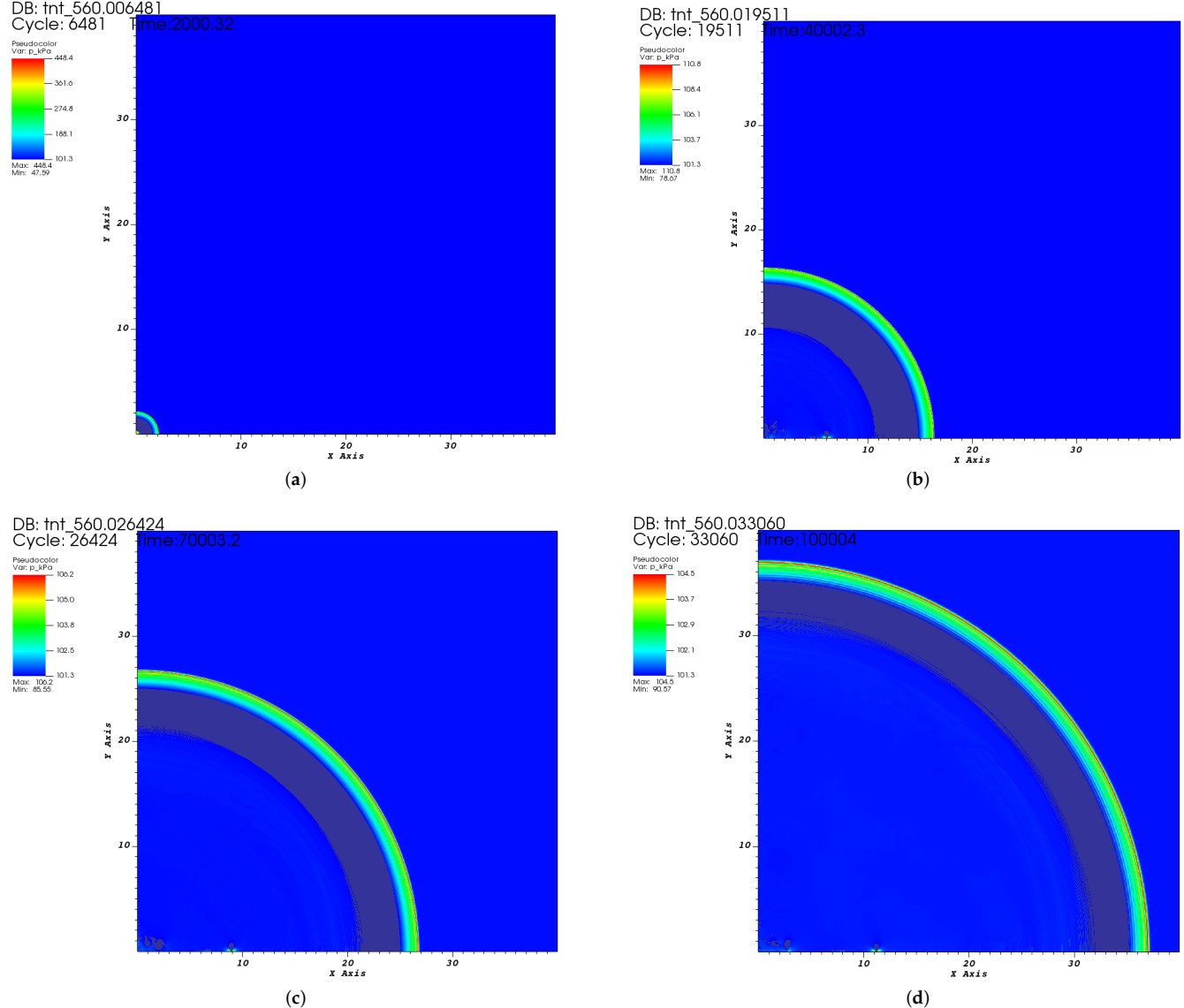

**Figure 2.** Snapshots of the pressure wave from the 0.90718 kg explosion of hemispherical TNT into air at atmospheric pressure (approximately 101.35 kPa): (**a**) 2 ms, (**b**) 40 ms, (**c**) 70 ms, and (**d**) 100 ms after programmed detonation.

Figure 4 shows the pressure waves recorded at 36.576 m (120 ft) from the source. Note that when compared to some of the larger pressure, early time pressure waves on the left-hand side of the plots in Figure 3, these waves are relatively smooth and have a shallower initial slope as they ramp up to maximum incident overpressure. This corresponds to the fact that at this distance the waves are no longer proper shock waves. In fact, the wave velocity is approximately that of the speed of sound in air. The area under the positive portion of the overpressure wave is the total incident impulse at this point. As will be shown in the next section, pressure time histories of this type are useful in that they can be used as boundary conditions in subsequent simulations to study the dynamic effects of realistic incident waves on structures.

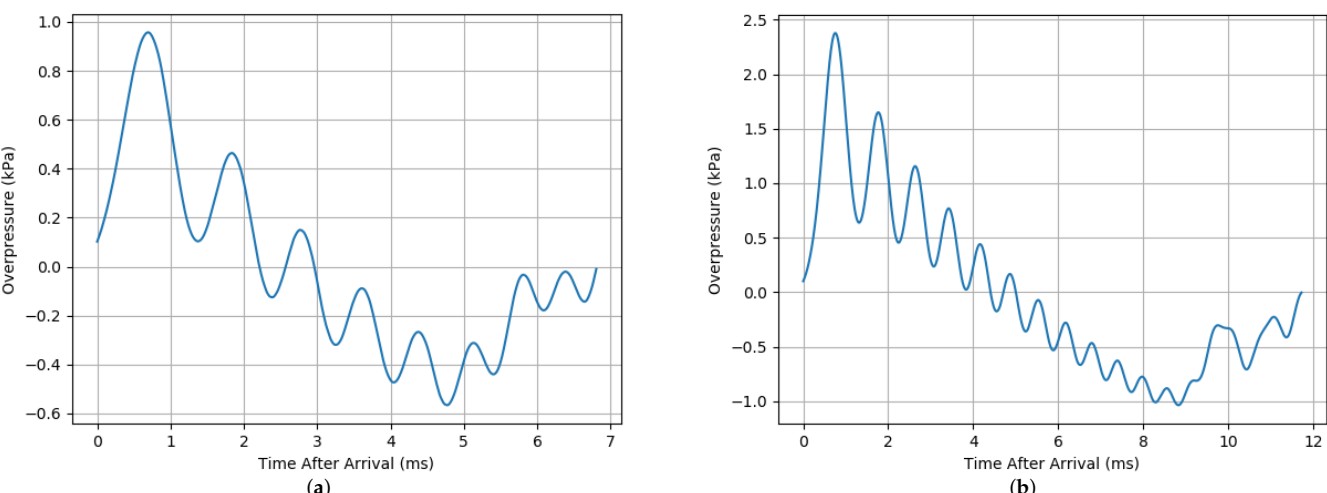

**Figure 3.** Pressure time histories from four simulations of hemispherical TNT detonation of different weights: (**a**) 0.06123 kg (0.135 lb), (**b**) 0.45359 kg (1 lb), (**c**) 0.90718 kg (2 lb), and (**d**) 1.81436 kg (4 lb).

**Figure 4.** *Cont.*

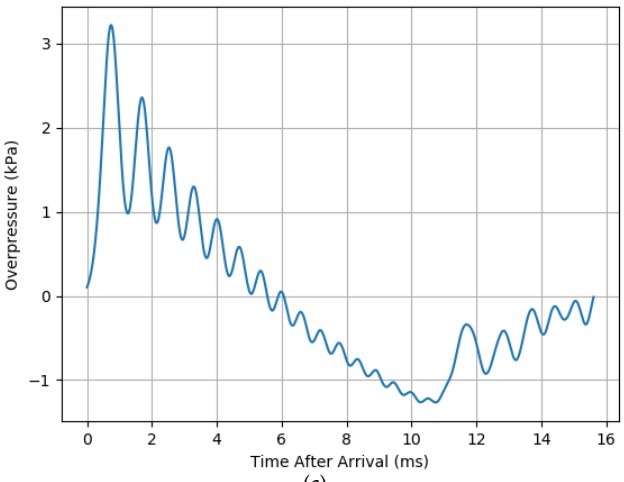

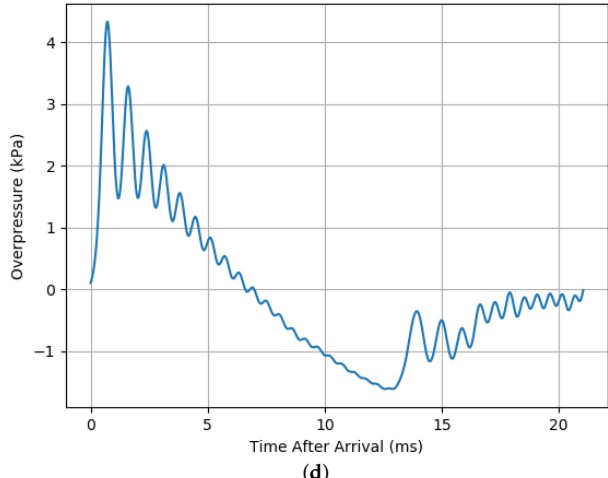

**Figure 4.** Pressure time histories from four simulations of hemispherical TNT detonation at 36.576 m (120 ft): (**a**) 0.06123 kg (0.135 lb), (**b**) 0.45359 kg (1 lb), (**c**) 0.90718 kg (2 lb), and (**d**) 1.81436 kg (4 lb).

Simulation of Detonation Cord

Further simulations were performed in order to study the variation of effects due to geometry. In particular 1.829 m of detonation cord suspended 1.524 m and parallel to the ground was detonated in a 4.5 m × 4 m × 3 m domain of air under atmospheric pressure (Figure 5). The cord is comprised of a 0.18 cm radius cylinder of TNT, so that ultimately 32.27 g is detonated. Pressure tracers are placed at regular distances from the center of the cord at a height of 1.524 m. Figure 5 shows snapshots of the resulting pressure waves in time.

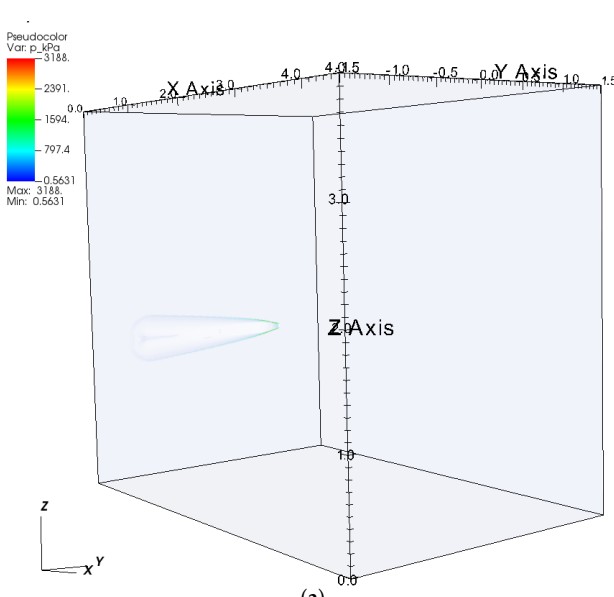

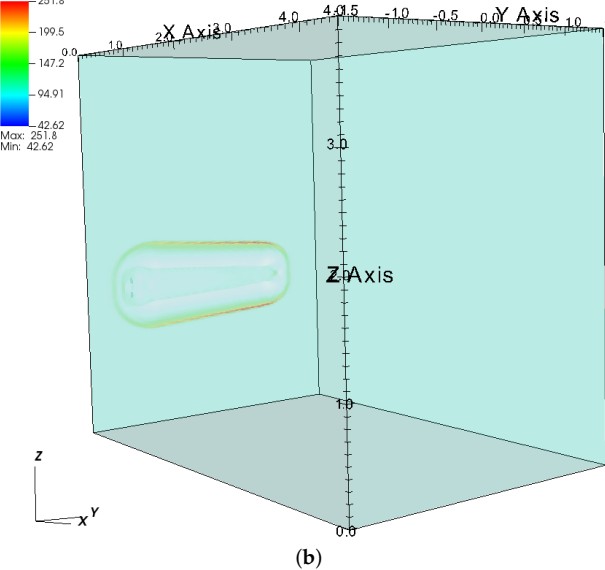

**Figure 5.** *Cont.*

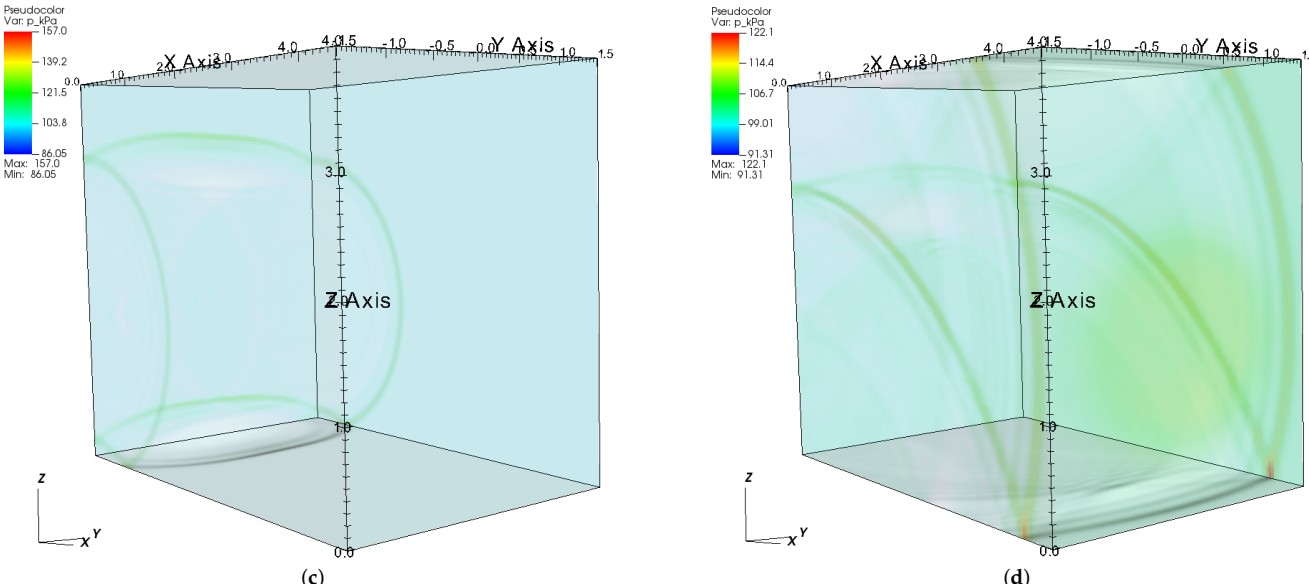

(c)                                     (d)

**Figure 5.** Snapshots of the pressure wave from explosion of a detonation cord (**a**) 0.27 ms, (**b**) 0.7 ms, (**c**) 4.6 ms, and (**d**) 12 ms after programmed detonation.

### 2.3. Blast-Barrier Interaction

Simulations of the interaction of incident overpressure waves and lightweight Lexan barriers were performed to gauge the effectiveness of simple modular structures to maintain "Public Withdrawal Distance" conditions where incident overpressures are already quite low. The 2D plane strain simulations were performed, given the assumption that multiple barriers could be placed alongside each other to minimize any edge effects. Further larger 3D cases of interest were explored to visualize and quantify the effects of lateral wraparound for standalone barriers.

Figure 6 presents a "cartoon" depiction of these simulations with labeled boundary conditions. Again the lower boundary is taken as a symmetry plane to estimate ground interactions as perfect reflections. The upper and outer boundaries have pressure continuous non-reflecting conditions. The $x = 0$ plane is given a pressure load curve corresponding to the pressure tracer time histories derived from the free-field blast simulations (Figure 4). It is assumed that in the far field the incident waves are planar. A problem arose in earlier simulations where reflections off of the barrier reached the $x = 0$ plane a re-reflected back into the problem domain before the relevant dynamic events could conclude, causing undesirable boundary effects. It was found that the non-reflecting boundary conditions did not coexist well with the pressure load curves and thus caused numerical issues with the incident pressure waves. To avoid these issues, the barrier was placed at a distance $d = \frac{1}{2}ct_{wave}$, where $c$ is the speed of sound in air (approximately 343 m/s) and $t_{wave}$ is the wavelength (in time) of the incident pressure wave, including positive and negative overpressure phases. Because the far-field waves are traveling at approximately the speed of sound, under these conditions the entire incident wave enters the domain before reflections can return to the boundary. Then, at time $t_{wave}$, the pressure load curve boundary conditions are replaced with pressure continuous, non-reflecting conditions which eliminate the problem of reflection. The 3D simulations were performed in half-symmetry, so that the $y = 0$ plane was a symmetry plane and the $y_{max}$ also had pressure continuous and non-reflecting conditions.

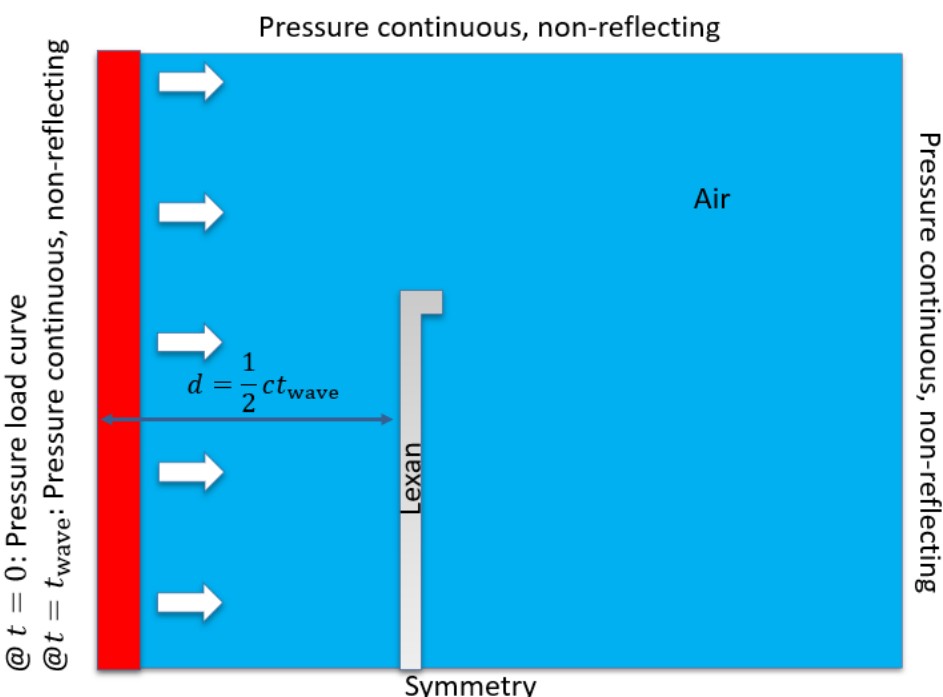

**Figure 6.** A "cartoon" depiction of the setup of the blast-structure interactions simulations with materials and boundary conditions labeled. (Not to scale)

Various simple designs of mitigation barriers were studied. These included three major types: single fairing, compound fairing, and deep-roof (Figure 7). The barriers are all 1.2 m wide, and 3.8 cm thick. The total height varies with the length and angle of the fairing, but the bases are approximately 2.2 m high.

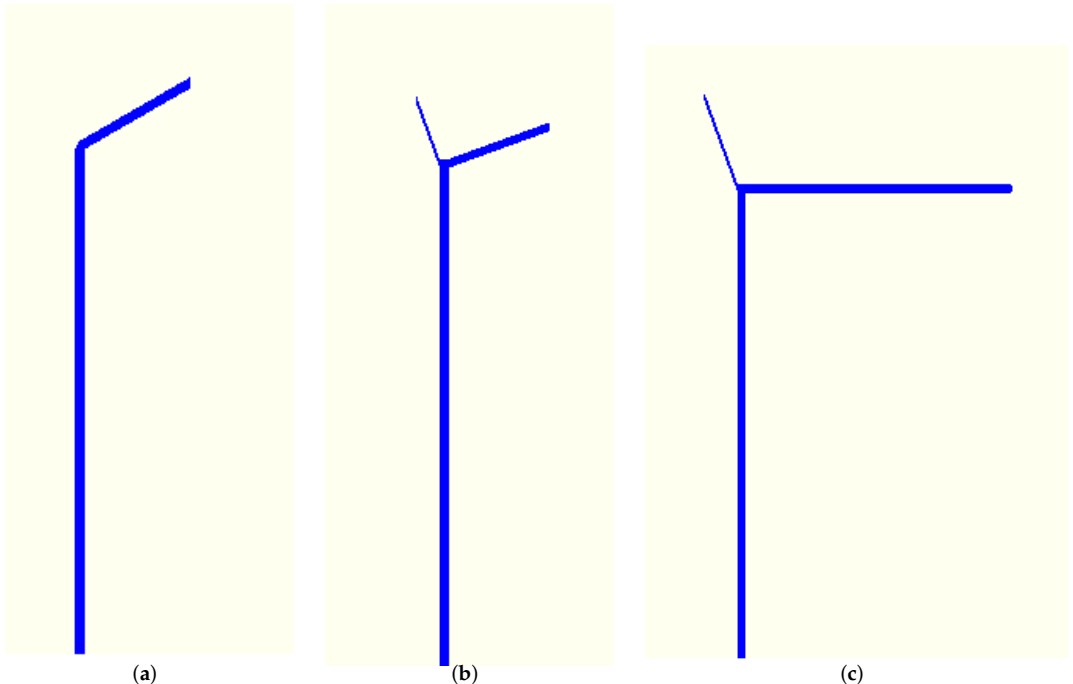

**Figure 7.** Representative cross-sections of the types of mitigation barriers studied: (**a**) Single-fairing barrier. (**b**) Compound-fairing barrier. (**c**) Barrier with deep roof.

All calculations utilized a graded mesh which was most refined in the area around the mitigation barrier. The 2D plane strain simulations ultimately contained around 1.2 million zones. The 3D simulations in general utilized a coarser mesh that was graded more aggressively, but still contained on the order of 10 million zones per simulation. Figures 8 and 9 show snapshops of the pressure fields in representative 2D and 3D simulations, respectively.

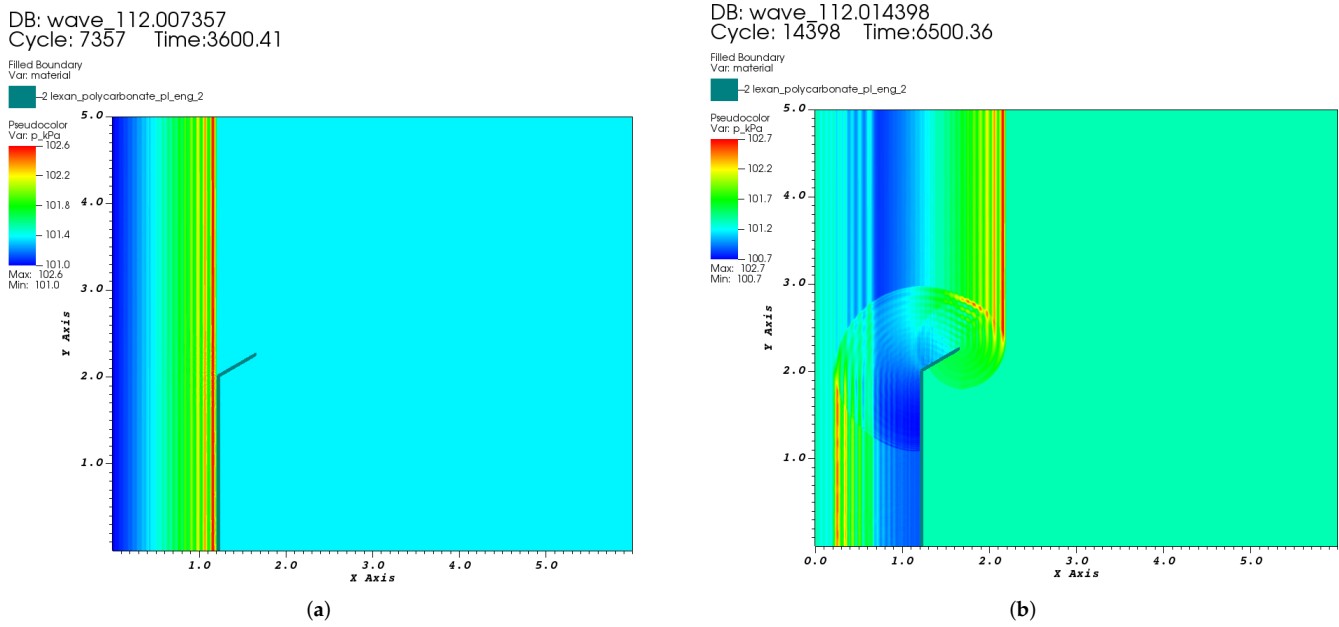

**Figure 8.** Images from a representative 2D plane strain blast-barrier mitigation simulation: (**a**) at arrival time of wave at barrier. (**b**) During dynamic interaction event.

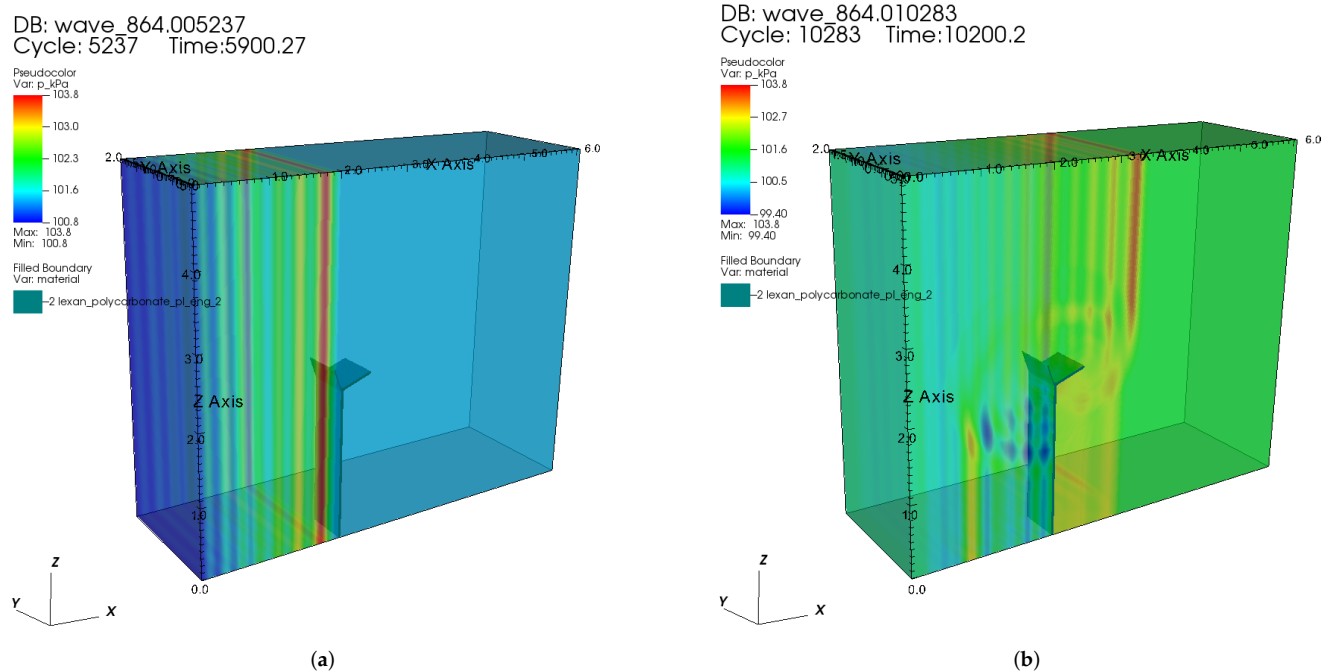

**Figure 9.** Images from a representative 3D blast-barrier mitigation simulation: (**a**) at arrival time of wave at barrier. (**b**) During dynamic interaction event.

## 3. Results

### 3.1. Comparison of Predicted and Simulated Blast Effects

Of particular interest is the comparison of numerical results with the classical Kingery-Blumash type empirical curves. The fits to these data are most conveniently given by Swisdak [4] in the following form:

$$\exp\left(A + B \ln K + C(\ln K)^2 + D(\ln K)^3 + E(\ln K)^4 + F(\ln K)^5 + G(\ln K)^6\right) \qquad (7)$$

Here, $K$ is the $K$ factor given by Equation (1). The curve parameters $A - G$ for Equation (7) fitting peak incident overpressure and positive impulse are given in Tables 4 and 5, respectively.

**Table 4.** Parameters for Equation (7) for peak incident overpressure (from Swisdak).

| $K$ Values | $A$ | $B$ | $C$ | $D$ | $E$ | $F$ | $G$ |
|---|---|---|---|---|---|---|---|
| 0.2–2.9 | 7.2106 | −2.1069 | −0.3229 | 0.1117 | 0.0685 | 0.0 | 0.0 |
| 2.9–23.8 | 7.5938 | −3.0523 | 0.40977 | 0.0261 | −0.01267 | 0.0 | 0.0 |
| 23.8–198.5 | 6.0536 | −1.4066 | 0.0 | 0.0 | 0.0 | 0.0 | 0.0 |

**Table 5.** Parameters for Equation (7) for incident impulse (from Swisdak).

| $K$ Values | $A$ | $B$ | $C$ | $D$ | $E$ | $F$ | $G$ |
|---|---|---|---|---|---|---|---|
| 0.2–0.96 | 5.522 | 1.117 | 0.6 | −0.292 | −0.087 | 0.0 | 0.0 |
| 0.96–2.38 | 5.465 | −0.308 | −1.464 | 1.362 | −0.432 | 0.0 | 0.0 |
| 2.38–33.7 | 5.2749 | −0.4677 | −0.2499 | 0.0588 | −0.00554 | 0.0 | 0.0 |
| 33.7–158.7 | 5.9825 | −1.062 | 0.0 | 0.0 | 0.0 | 0.0 | 0.0 |

The results for blast overpressure are also compared with predictions from the Taylor–von Neumann–Sedov result. It is shown in [31] that from this solution, the blast radius and corresponding peak pressure are given as a function of time as:

$$R(t) = \beta \left(\frac{E t^2}{\rho_0}\right)^{1/5} \qquad (8)$$

$$p(t) = \frac{2}{\gamma + 1} \rho_0 \left(\frac{2}{5} \frac{R}{t}\right)^2 \qquad (9)$$

Here, $E$ is the energy of the explosion, $\rho_0$ the initial density of the air. $\gamma$ is the same parameter appearing in Equation (3), and $\beta$ is a corresponding parameter which has a value of 1.033 for air. Solving (8) for $t$ and substituting into (9) yields an equation for pressure as a function of blast radius:

$$p(R) = \frac{8}{25(\gamma + 1)} E R^{-3} \beta^5 \qquad (10)$$

This result is valid for a point source explosion in a zero-pressure medium expanding spherically from the origin. In order to compare with our hemispherical results, we compare to a blast having twice the energy of 1 kg TNT; this corresponds with the fact that the symmetry conditions on the floor of our free-field simulations make them numerically equivalent to spherical blasts of the same radius, i.e., twice the weight.

Figure 10 shows the comparisons for peak incident blast overpressure of the free field hemispherical and detonation cord simulations with Equations (7) and (10). Figure 11 shows the corresponding positive impulses calculated from the pressure tracers by numerically integrating the positive portions of the pressure tracers from the hemispherical simulations compared to Equation (7).

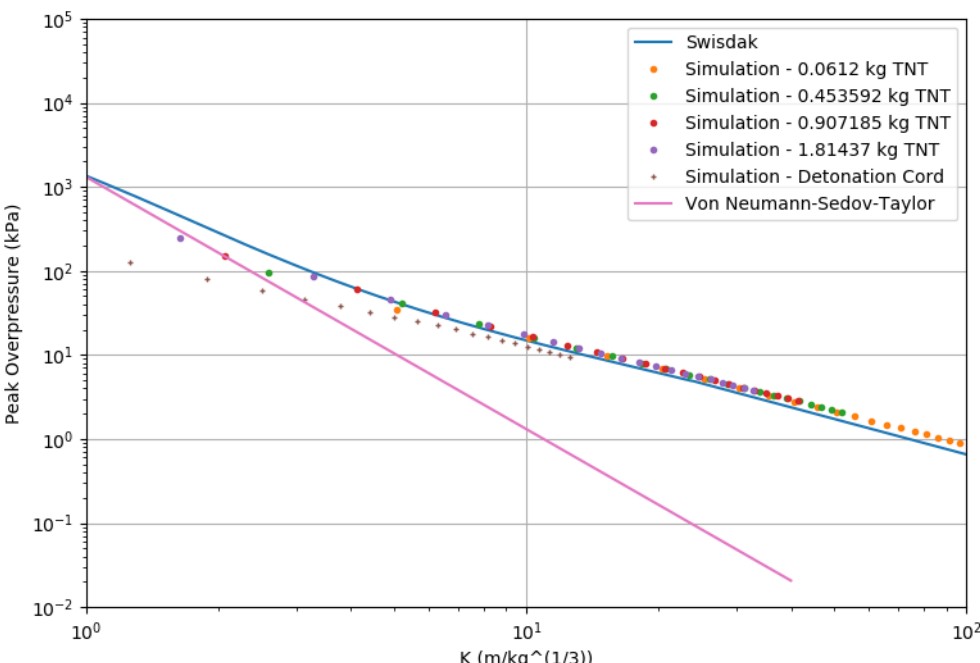

**Figure 10.** Peak incident blast overpressure versus *K* factor from the four TNT hemispherical simulations and the detonation cord simulation in comparison with the KB curve from Swisdak and the Taylor–von Neumann–Sedov prediction.

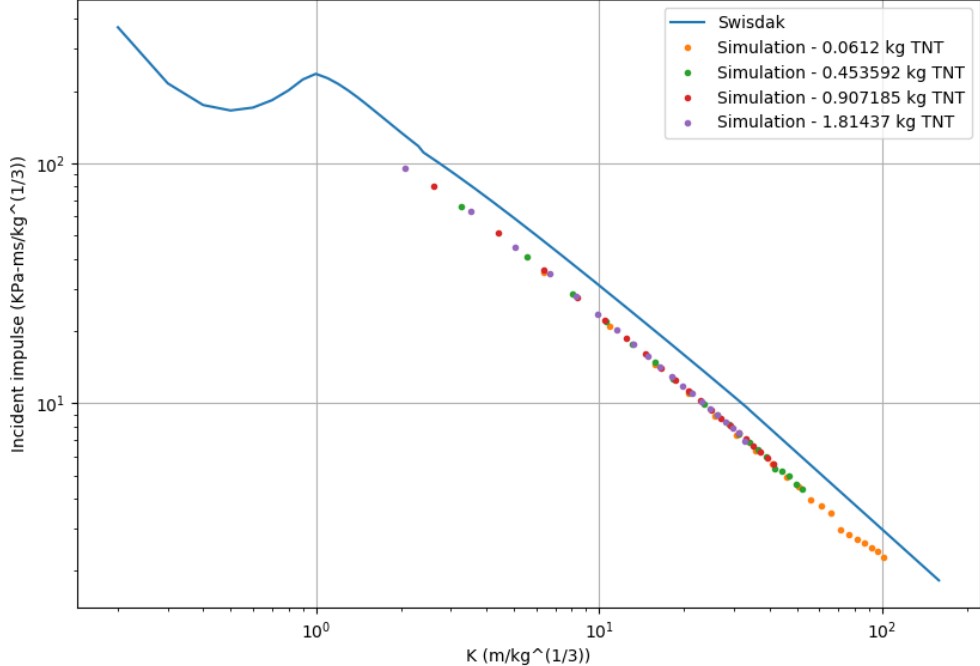

**Figure 11.** Positive impulse versus *K* factor from the four TNT hemispherical simulations in comparison with the KB curve from Swisdak.

### 3.2. Mitigation Effectiveness

The effectiveness of the simple Lexan barriers at mitigating incident pressure fields is investigated with particular emphasis on the so called "Public Withdrawal Distance". In m/kg$^{1/3}$ units this corresponds to a *K* value of 130.12 and a blast overpressure of approximately 0.4516 kPa (0.0655 psi). To gauge mitigation effectiveness, pressure tracers were placed in a uniform grid behind the barriers in the present simulations; the pressure

time histories are then queried based on the aforementioned peak pressure criterion, and a "bubble" of space satisfying the maximum desired conditions can be plotted.

Figures 12 and 13 show the analysis of a single fairing mitigation barrier interacting with a wave from 0.06123 kg (0.135 lb) of TNT at approximately 36.576 meters (120 ft). The peak incident overpressure in this case is approximately 0.95 kPa (0.137 psi). Figure 12b shows that in the plane strain case, the pressure is effectively mitigated behind the barrier below 0.4516 kPa for a region over 2 m high and extent of almost 5 m. Figure 13b shows than in the 3D case with a barrier of finite width, there are small localized regions near the edges, center, and ground where edge wraparound and reflections exceed this pressure threshold.

Figures 14 and 15 show the results of plane strain analysis from a 0.45359 kg (1 lb) TNT charge at approximately 36.576 m (120 ft) interacting with a compound fairing and 'deep roof' type barrier. The peak incident overpressures in this case is approximately 2.38 kPa (0.345 psi). In both cases, the incident pressure wave is partially mitigated, so there are still large regions behind the barrier seeing pressures larger than 0.4516 kPa. The 'deep roof' style barrier provides a large 'bubble' for pressures under 0.4516 kPa. It is worth noting that in all cases, the largest pressures behind the barrier occur when the wave which passes over the top reflects back off the ground and the back of the barrier. The incident wave over the barrier has been mitigated below the target pressure, but the reflections exceed it.

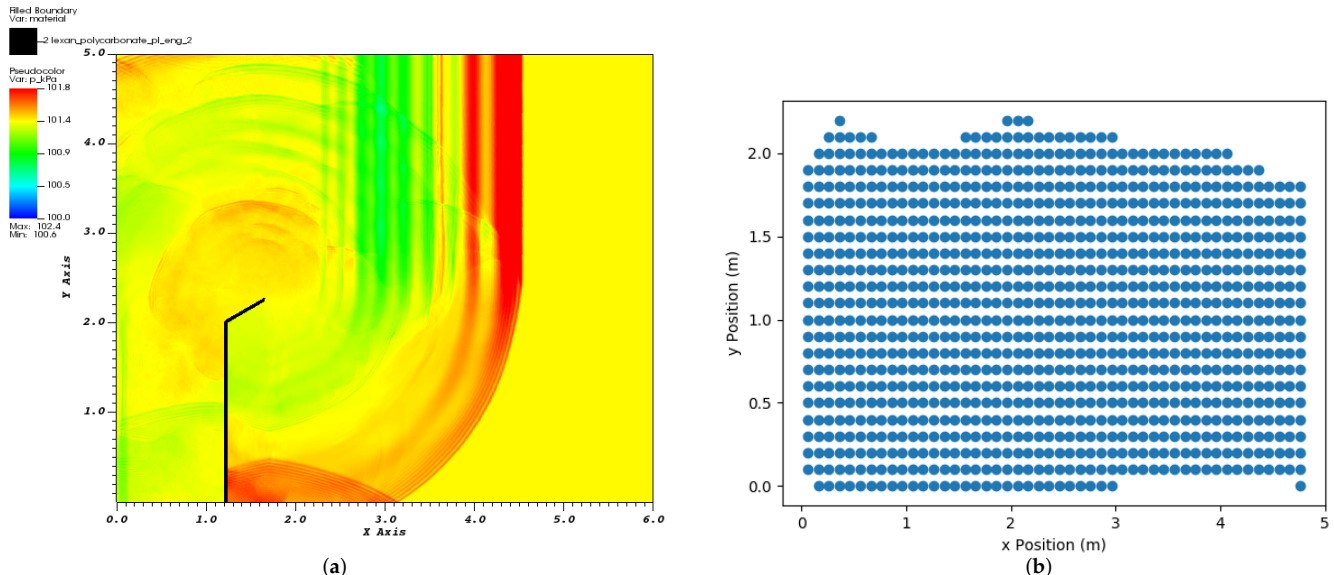

(**a**)

(**b**)

**Figure 12.** Results from a plane strain analysis of a single-fairing mitigation barrier loaded by a wave generated from 0.06123 kg of TNT at approximately 36.576 m. (**a**) A snapshot of the wave reflecting over the barrier. The color gradient is set so that max (red) values are above the 0.4516 kPa overpressure threshold. (**b**) The "bubble" behind the barrier for which max overpressure was beneath 0.4516 kPa.

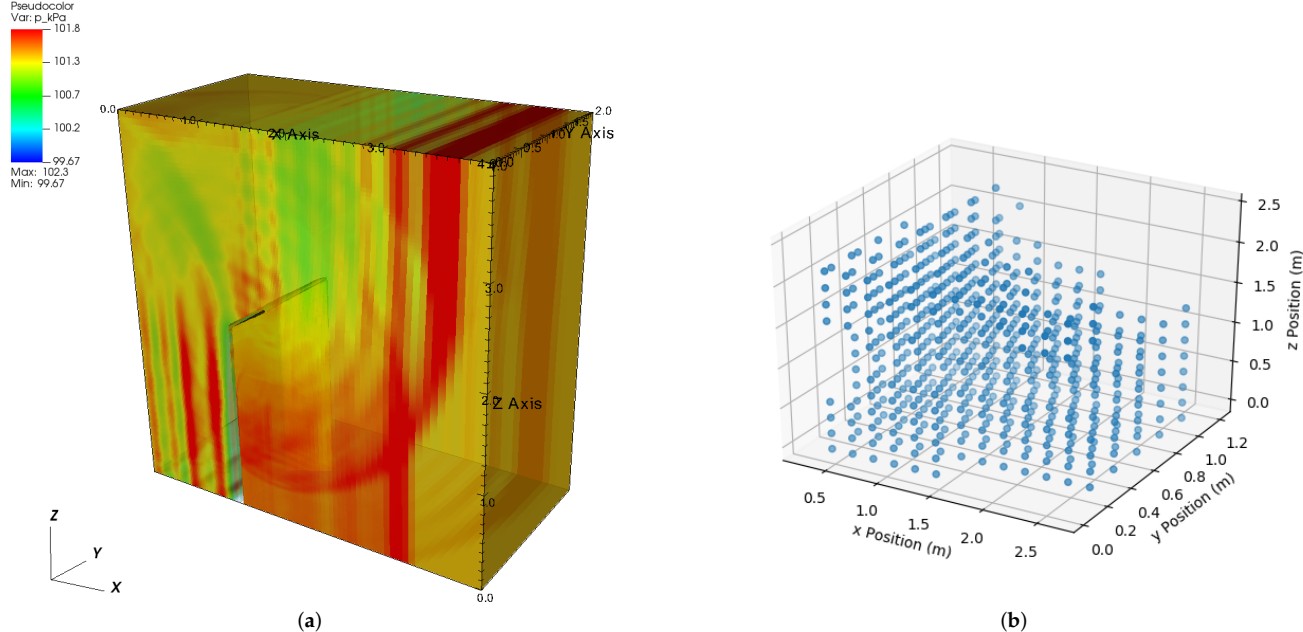

(**a**)

(**b**)

**Figure 13.** Results from a 3D analysis of a single-fairing mitigation barrier loaded by a wave generated from 0.06123 kg of TNT at approximately 36.576 m. (**a**) A snapshot of the wave reflecting over the barrier. The color gradient is set so that max (red) values are above the 0.4516 kPa overpressure threshold. (**b**) The "bubble" behind the barrier for which max overpressure was beneath 0.4516 kPa.

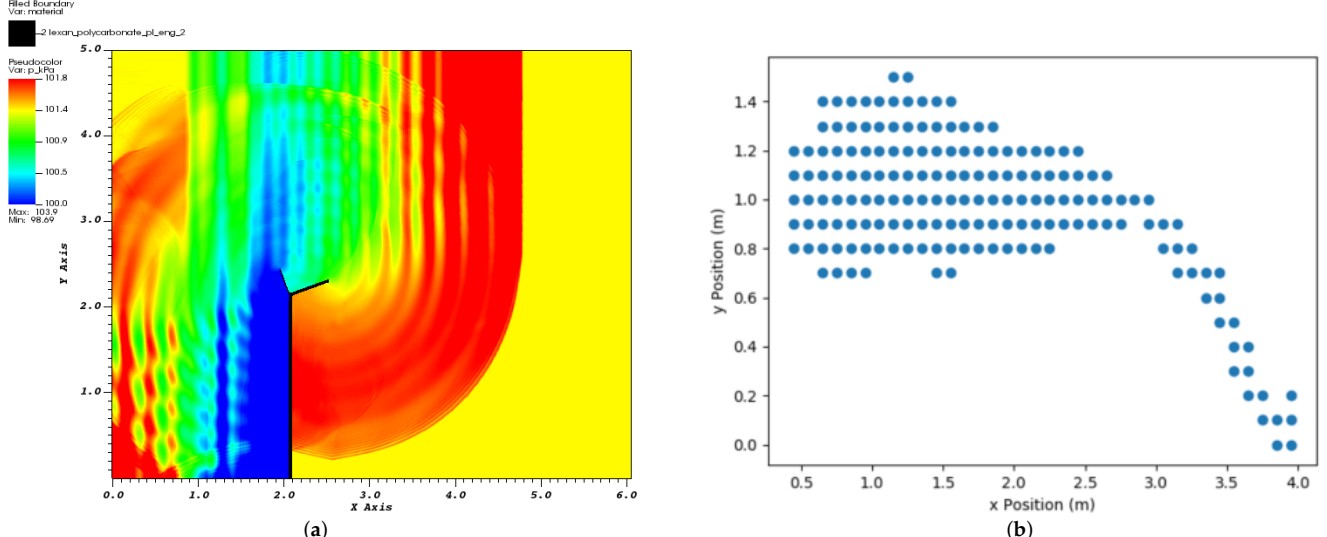

(**a**)

(**b**)

**Figure 14.** Results from a plane strain analysis of a compound-fairing mitigation barrier loaded by a wave generated from 0.45359 kg of TNT at approximately 36.576 m. (**a**) A snapshot of the wave reflecting over the barrier. The color gradient is set so that max (red) values are above the 0.4516 kPa overpressure threshold. (**b**) The "bubble" behind the barrier for which max overpressure was beneath 0.4516 kPa.

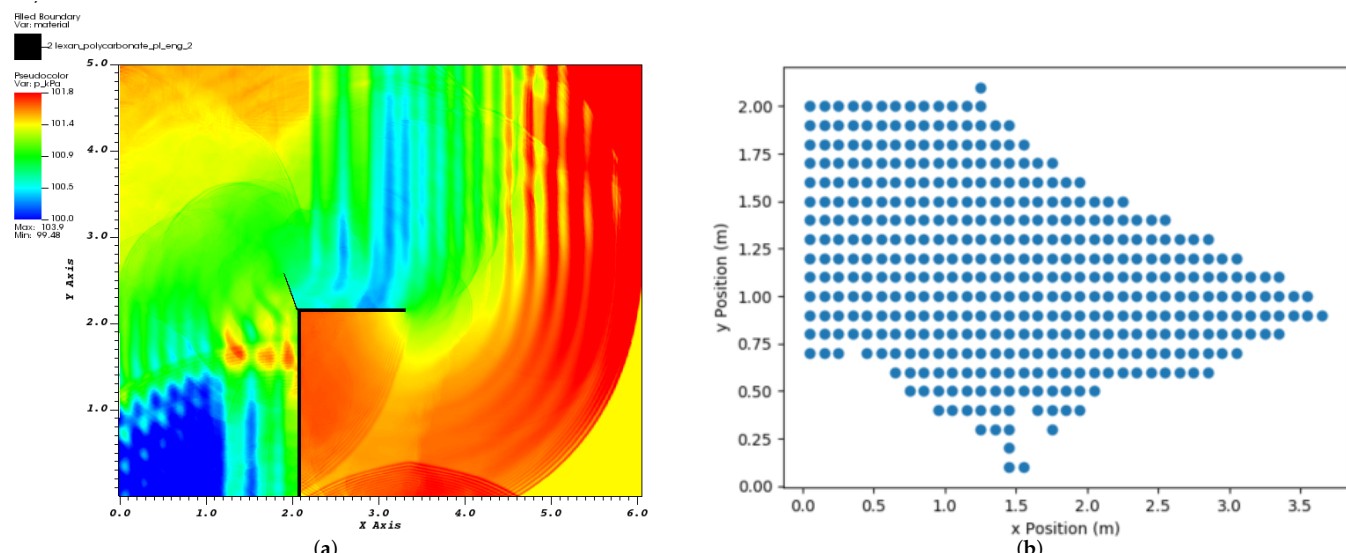

**Figure 15.** Results from a plane strain analysis of a 'deep-roof' type mitigation barrier loaded by a wave generated from 0.45359 kg of TNT at approximately 36.576 m. (**a**) A snapshot of the wave reflecting over the barrier. The color gradient is set so that max (red) values are above the 0.4516 kPa overpressure threshold. (**b**) The "bubble" behind the barrier for which max overpressure was beneath 0.4516 kPa.

## 4. Discussion

The calculated incident peak overpressures from the free-field hemispherical blast simulations show good agreement with the KB predictions in Figure 10. Interestingly, the largest deviation from the KB curve appears to at the points closest to the explosions yielding the largest overpressure; these first three points lie closer to the von-Neumann-Taylor-Sedov prediction, which rapidly deviates from the Swisdak (KB) curve. The analytical prediction is only valid at an intermediate distance from large explosions; it breaks down near the explosion, as the point source assumption washes out details of the actual detonation event, but also in the very far-field, where the assumption that $p_0 = 0$ in the ambient gas begins to corrupt the results as the incident overpressure approaches the ambient atmospheric pressure. Since the deviation of the data points from the empirical curve is likely within the experimental errors of the original fits, the fact that the data seem to jump from the analytical to the empirical curves may be coincidental. In the very far-field, the KB predictions seem to be doing a reasonable job at predicting the calculated overpressures, despite the fact that it is fit to data from explosions that were orders of magnitude larger.

As expected, the KB predictions do not do well at predicting the overpressures near the detonation cord. The asymmetrical blast wave from a long, thin cylindrical cord lit at one end reaches a nearby point at different times, making the peak pressure smaller than that predicted from a localized (hemispherical) source. However, with greater distance this time delay becomes smaller and the data appears to converge onto the KB curve.

There is a larger discrepancy between the KB-predicted and calculated incident impulses in Figure 11. There is very good agreement in the slope of the data versus the curve, but the free-field simulations appear to uniformly under-predict the impulse relative to the KB curve by a relatively small amount. Given the better agreement in the peak overpressures, there may be some discrepancy in the shape or duration of the whole incident pressure wave. The source of this error could be numerical or physical. There may be low pressure effects to the waves which we not captured in the very large Kingery tests. Note the oscillations that appear in the smaller pressure time histories in Figure 3. These appear after the sharp shock-like pressure spikes decay into more smooth waves traveling at sound speed. While these oscillations could be numerical effects, subsequent calculations were

conducted to investigate this by changing mesh size and the position of the tracer nodes, which appeared to have no effects on the oscillations at distance. Thus it is possible that the oscillations in the far-field small pressure waves is in fact a physical phenomenon. This could in part explain the discrepancy in impulse when there is good agreement in the peak pressure magnitude. It is noted that similar but less pronounced oscillations also seem to appear in the farthest-field pressure histories in the works of Xue et al., Ding et al. [20,22]. There do not appear to be other curves available from similarly small charges at distance to compare with Figure 4. Taking experimental measurements of incident pressures and impulses much smaller that atmospheric pressure very far away from small explosives is quite difficult.

It is further noted that the overall predictive accuracy in the subsequent blast-mitigation dynamic simulations is in part dependent on the accuracy of the predicted incident waves. At 36.576 m (120 ft), only the wave from the smallest (0.06123 kg) charge was mitigated down below the "Public Withdrawal Distance" value of 0.4516 kPa consistency behind the barrier under plane strain conditions; a large bubble was confirmed under a larger 3D simulation of a single barrier, though there were small regions near the edge and center where pressure rose higher in this case. The practical suggestion gleaned from this is that when implementing this type of barrier it may be wise to include more than one side by side to approximate the plane strain condition.

It is noted that the relevant hydrodynamics effects are likely more accurately captured in the 3D simulations. For example, mixing and turbulence are fundamentally 3D phenomena. Furthermore, resolution of any smaller-scale effects is inherently limited by the resolution of the simulation at those scales. However, due to the relatively low velocities and pressures these factors are not thought to have much influence in the cases studies here. Recent work has shown that purposefully exploiting wave interference can be useful in blast mitigation for incident strong shocks [32].

Finally, the effectiveness of using TNT equivalence values to compare expected blast effects from different explosives depends on the situation [33]. The present work has employed only a simple model of TNT with the simplest numerical detonation/burn assumptions. This seemed appropriate when gauging effects in the far-field, when the incident waves are sufficiently decoupled from the nuances of the blast and the blast products. Further work should be conducted to verify the accuracy of the KB charts and the predictions made here with other types of explosive, as well as to simulations with more sophisticated burn models (e.g., ignition and growth [34]).

## 5. Conclusions

The present manuscript lays out two open problems (namely, what incident overpressure and impulses are felt at given distances from relatively small hemispherical ground charges, and how well can certain types of boundaries mitigate the incident overpressure below a certain threshold). It then describes the results of numerical investigations to attempt to answer these questions. A major motivating factor in this research is the uncertainty in the available empirical curve fits (e.g., Kingery–Bulmash). The source of this uncertainty is twofold: there is relatively large error between some of the original data and the available fitting curves, and the original data were taken for explosions that were many orders of magnitude larger than the charges investigated here. A major unknown remains the extent to which the assumed scaling described by Equation 1 (distance by the cubed root of charge weight) holds as weights become small. The free-field blast simulations presented here indicate that the strong shock of the initial blast smoothed out within the distance simulated and continued to propagate near the sound speed. The slowing of the wave speed is in fact predicted by the empirical Swisdak (KB) equations, but there remains uncertainty into how this change in the physics regime and the shape of the waveform effects the ultimate impulse at different scales. A benefit of the direct numerical calculations is the availability of the full waveforms in time at all distances in the simulation domain; this was further leveraged in the subsequent mitigation simulations. The ultimate shape of

the incident pressure wave may be another degree of freedom which is not fully captured by the *K* factor scaling. This may explain why the simulations agree well with the peak overpressure and the slope of the impulse curves from Swisdak, but seem to consistently predict slightly smaller impulse magnitudes.

It was never assumed that the empirical curves would or should be "exact" predictors of incident overpressure and impulse for a given case. While this was a primary motivator for the present attempts for a direct physics-based prediction, it is also not assumed that these predictions will correspond exactly to any field case. Ultimately, the analyst, engineer, or responsible person must weigh uncertainty and risk to assess a given scenario. It is hoped that the present simulations (or others like them) could be used in uncertainty quantification efforts for blasts effects in wider-varying scenarios.

The blast-barrier mitigation simulations presented here were also motivated by this desire to mitigate risk and uphold safety standards. The "Public Withdrawal Distance" or "K328" threshold was taken as a more-or-less arbitrary datum against which to gauge effectiveness. The findings of this work should not be used to indicate whether a given scenario is "safe", but rather to elucidate some of the physical mechanisms of mitigation in a dynamic blast event. Safety standards and acceptable risk vary from scenario to scenario; this work provides a methodology of analyzing the effectiveness of hypothetical tools to decrease risk.

The specific barrier designs presented here were somewhat ad hoc and experimental. The fairings were designed to reflect incident waves and further mitigate overpressure from wraparound over the top. The double fairing was intended to facilitate mitigation further by partially reflecting the incident wave from the backward-facing fairing. The "deep roof" concept was designed to provide even further mitigation. Each subsequent design was found to enhance mitigation. The results indicate that these types of simple barriers are in fact effective at mitigating incident pressure and impulse. They do not, however, eliminate these risks. Ultimately, distance from the source is the surest form of mitigation.

All conclusions herein would be much strengthened by specific field test data taken from experiments with the same charge weights and at the same distances, both in the free-field and behind the proposed barriers. As far as the author knows, no data exists that is a direct match for the scenarios described here. Currently available state-of-the-art instrumentation may be able to reliably measure the small dynamic pressures considered in this study. The experimental verification of these scenarios is outside of the scope of the present work.

**Funding:** This research received no external funding.

**Data Availability Statement:** Not applicable.

**Acknowledgments:** This work was performed under the auspices of the U.S. Department of Energy by Lawrence Livermore National Laboratory under Contract DE-AC52-07NA27344. Special thanks to Mark M. Hart who posed the present problem and provided crucial expertise, intuition, and feedback. Further Thanks to Larry D. McMichael for guidance and expertise in setting up the numerical simulations.

**Conflicts of Interest:** The authors declare no conflict of interest.

## Abbreviations

The following abbreviations are used in this manuscript:

| | |
|---|---|
| 1D | One-dimensional |
| 2D | Two-dimensional |
| 3D | Three-dimensional |
| ALE | Arbitrary Lagrangian/Eulerian |
| KB | Kingery–Bulmash |

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
