# Peer review of "Numerical Analysis of Blast Effects and Mitigation in the Far-Field from Small Explosions"

_applsci, doi:10.3390/app12178824_

Round 1

Reviewer 1 Report

The manuscript entitled "Numerical analysis of blast effects and mitigation in the far-field from small explosions" has been reviewed. After carefully reading this paper, my suggestion is major revison, and some issues should be incorporated.

1. More descriptions and comparisons of the differences of blast effects from big and small explosions should be added in the Introduction Section. After this, the significance of this study can be clear.

2. Why the net explosive weights under three barrier conditions in Section 3.2 are different?

3. Why these three kinds of barriers should be analyzed in this article?

4. A conclusion section should be added, and the difference between the analysis results with previous results should be carefully introduced.

5. Both 2D and 3D simulations can be found in this article, and no clear descriptions were shown, which may confuse readers.

6. The layout of the article should be revised, and an independent section namely Simulation results calibration was recommended to be added. Meanwhile, more results verifications should be added. 

Reviewer 2 Report

Dear Adam G Taylor,

The paper's numerical analysis of blast effects and Mitigation in the far-field from the small explosion is an interesting work on the subject of numerical hydrocode analysis. This paper presents in a significant way the historical research as well as problems that appear concerning KB model. Usually, in some of the national standards, it can be found empirical equation which allows for establishing the theoretical AB value and how it’s differ with distance, however, this is a big simplification of what can be concluded from this work. On the merits, the paper is well written. What I found inconvenient was only the editorial side that concerned figures. The description of Figures 3, 4, and 5 was before the figure. Moreover, in the founding section, there is additional irrelevant information.

Best regards

Reviewer

Author Response

The author thanks the reviewer for the helpful summary and critique. The issues with the figure placement will be address in the revised document, and the text will be revised.

Thanks again,

-Adam G Taylor

Reviewer 3 Report

Manuscript does not present a good hypothesis.

Title and Abstract does not completely correspond the subject.

The title and abstract should be more specific, highlighting the main point.

“K-factor” is basically scaled distance (blasting term).

In introduction it is not clear is the charge free or in the ground, as well as “blast effects” are they air blast or ground vibration velocity.

Numbers on almost all figures are too small, unreadable.

Due to poorly chosen words, some sentences are unclear and confusing, English need some rework.

There are some misprints within manuscript, example “(1D)” in line 58.

What does “easily available pressure gauges” means. There are instruments for measuring air blast in the market.

ALE3D hydrocode should be explained in more detail.

Figures 10 and 11 should be after textual part where they are mentioned (after line 237)

Simulation should be supported with field measurements.

There is no real conclusion of manuscript (as a title as well as text). No explanation about applicability in the field. What is the real benefit of using described analysis? The novelty should be highlighted in the title, abstract, conclusion and throughout whole manuscript.

Reviewer 4 Report

The topic is quite interesting and challenging. The authors achieve an impressive result and made a perfect summary. However, I wish to point out several items below to improve the paper.

- It is difficult to read X and Y axis, and contour in the Figures which contain numerical analysis results. I think it would look better with a larger font.

- In fig. 13, It is difficult to see values because the values are overlapped each other.

- There are some typos including “Figure 12(b) In the plane strain case, the ….” In line 248.

- In this journal, the conclusion section is an option. But I recommend adding a concluding section with a summary of this research for readers (it is an option).

Author Response

The author thanks you for the positive feedback and constructive criticism. The issues with the figures (readability, etc.) will be addresses in the revised document. The document will be further revised for typos; thanks for pointing one out. There will also be included a new conclusion section summarizing the work.

Thanks again.

-Adam G Taylor